# Patient perspectives of the Self-management and Educational Technology tool for Atrial Fibrillation (SETAF): A mixed-methods study in Singapore

Jennifer Nathania[1], Brigitte Fong Yeong Woo[2]*, Boon Piang Cher[3,4], Kai Yee Toh[3,4], Wei-Yan Aloysius Chia[3,4], Yee Wei Lim[1,5], Hubertus J. M. Vrijhoef[6,7], Toon Wei Lim[1,8]

1 Department of Medicine, Yong Loo Lin School of Medicine, National University of Singapore, Singapore, Singapore, 2 Alice Lee Centre for Nursing Studies, Yong Loo Lin School of Medicine, National University of Singapore, Singapore, Singapore, 3 Centre for Health Services and Policy Research, Saw Swee Hock School of Public Health, National University of Singapore, Singapore, Singapore, 4 National University Health System, Singapore, Singapore, 5 Saw Swee Hock School of Public Health, National University of Singapore, Singapore, Singapore, 6 Panaxea, Amsterdam, The Netherlands, 7 Department of Patient and Care, Maastricht University Medical Centre, Maastricht, The Netherlands, 8 Department of Cardiology, National University Hospital, Singapore, Singapore

* brigittewoo@nus.edu.sg

**Data Availability Statement:** Anonymized transcripts of all participant interviews are publicly available at 10.6084/m9.figshare.17294153.

## Abstract

### Background

Atrial fibrillation (AF) is the most common arrythmia and is associated with costly morbidity such as stroke and heart failure. Mobile health (mHealth) has potential to help bridge the gaps of traditional healthcare models that may be poorly suited to the sporadic nature of AF. The Self-management and Educational technology support Tool for AF patients (SETAF) was designed based on the preferences and needs of AF patients but more study is required to assess the acceptance of this novel tool.

### Objective

Explore the usability and acceptance of SETAF among AF patients in Singapore.

### Methods

A mixed methods study was conducted with AF patients who were purposively sampled from an outpatient cardiology clinic in Singapore. After 6 weeks of using SETAF, semi-structured interviews were performed, and data were analyzed inductively following a thematic analysis approach. Results from a short 4-item survey and application usage data were also analyzed descriptively. Both qualitative and quantitative results were organized and presented following the Technology Acceptance Model (TAM) framework.

### Results

A total of 37 patients participated in the study and 19 were interviewed. Participants perceived SETAF as useful for improving AF knowledge, self-management and access to

**Funding:** This study was funded by a Health Services Research New Investigator Grant (HSRNIG13nov002) from the National Medical Research Council, Singapore. The funders had no role in study design, data collection and analysis, decision to publish, or preparation of the manuscript.

**Competing interests:** The authors have declared that no competing interests exist.

**Abbreviations:** AF, Atrial fibrillation; ECG, Electrocardiogram; mHealth, Mobile health; PDA, Personal digital assistant; SETAF, Self-management and educational technology support tool for atrial fibrillation; SRQR, Standards for Reporting Qualitative Research; TAM, Technology acceptance model.

healthcare providers and was easy to use due to the guided tutorial and user-friendly interface. They also identified the need for better personalization of content, psychosocial support features and reduction of language barriers. Application usage data revealed preference for AF related content and decreased interaction with the motivational message component of SETAF over time. Overall, most of the participants would continue using SETAF and were willing to pay for it.

## Conclusions

AF patients in Singapore found SETAF useful and acceptable as a tool for AF management. The insights from this study not only support the potential of mHealth but may also inform the design and implementation of future mHealth tools for AF patients.

## Introduction

Due to the global ubiquity of mobile phones, with more than six billion users worldwide [1], mobile phones and other mobile devices have been increasingly used to deliver health interventions [2]. Singapore is reported to have one of the highest mobile phone penetrations in the world (157.7% in June 2021) [3]. In 2020, an estimated 88.43% of its population actively uses smartphones and is projected to increase over the years [4]. Mobile health (mHealth) is defined as the use of mobile devices including mobile phones, patient monitoring devices and personal digital assistants (PDAs) for medical and public health purposes [5]. A survey study, conducted with patients at a local hospital in Singapore, revealed positive attitudes towards mHealth, most of the patients were keen to use mHealth in the future and believed that mHealth could help to better manage their conditions [5]. Due to its widespread internet connectivity and high levels of mobile device use, Singapore is an ideal setting to introduce mHealth interventions and capitalize on its potential to improve health outcomes.

Atrial fibrillation (AF) is the most common cardiac arrythmia and is associated with costly cardiovascular morbidity such as stroke and heart failure [6–8]. The need for long term adherence to complex therapies, fluctuating symptoms and increased risk of adverse outcomes make AF a challenging condition for patients to manage [9]. Due to the sporadic and highly variable nature of disease, traditional healthcare models of pre-scheduled appointments with health providers may be poorly suited to meet the unpredictable and potentially urgent needs of AF patients [10]. With greater awareness, ageing populations and the rise in predisposing risk factors such as hypertension and diabetes mellitus, the prevalence of AF and its associated healthcare costs are expected to increase substantially [6, 7]. This highlights the importance of adopting new strategies to optimize the management of this growing epidemic.

mHealth technology has the potential to help address these issues by remote monitoring, patient education, promoting adherence to medications and healthier lifestyle behaviors. There is growing evidence of the positive effects from use of mHealth in the management of AF which can increase quality of life, AF knowledge, medication adherence and decreased emergency department visits [11, 12]. However, it is important to consider the contextual background of patient preferences, usability and acceptance of such technology in order to overcome barriers to use and promote medium to long term user engagement. In a previous study reporting the development phase of the Self-management and Educational technology support Tool for AF patients (SETAF), Cher et al. explored patients' and clinicians'

perspectives and attitudes towards the proposed mHealth intervention based on a demonstration prototype [13]. The insights gleaned from this study informed the design of SETAF to better address the needs and preferences of the local population. The current pilot study aimed to build on these findings to explore the usability and acceptability of SETAF among AF patients in Singapore.

In our preceding developmental study [13], we used a modified Technology Acceptance Model (TAM) for use in a healthcare context, but because there was no actual use of the mHealth tool, an evaluation of user experience was not possible. In the present study, we also drew on the TAM to examine user experience as the participants can now provide insights into the tool's perceived usefulness and perceived ease of use as originally described by Davis et al. [14]. These factors are proposed to influence the individual's intent to use the tool, and ultimately the actual usage behavior. Hence, the interaction between these key variables will be explored in our study.

## Methods

### Study design

This study used qualitative description to explore patients' experience in using SETAF to manage their AF condition. Qualitative description was preferred as it presents data from the participant's point of view and avoids interpretation through the lens of the researcher [15]. This approach is known to be useful for collecting data to develop and refine an intervention [16]. The 21 item Standards for Reporting Qualitative Research (SRQR) was used to guide the reporting of this study (S2 File) [17].

### Setting and sample

SETAF is an application integrated within the Philips Motiva system to complement future integrated care services for AF patients in Singapore through patient education, monitoring and self-management. Through SETAF, patients received motivational messages, health related surveys and self-management triage assessments. In addition, participants had access to educational videos and patient information sheets. The decision algorithm for the self-management triage and activity calendar for the SETAF application can be found as S1 Fig. The paired blood pressure and heart rate monitor automatically uploaded all measurements into the Motiva cloud database and enabled both patients and their care team to view the history and trends. The complete features of SETAF are described in Table 1.

At the beginning of the study, each participant was provided a tablet device using Android software and blood pressure monitor from the Philips Motiva system to access the SETAF

**Table 1. Components and features of SETAF.**

| Component | Features |
|---|---|
| Patient education | Educational videos and patient information sheets about atrial fibrillation and general health |
| | Health related surveys to test patient comprehension and compliance |
| Self-management | Actionable feedback about vital sign measurements |
| | Reminders and motivational messages to encourage healthy lifestyle choices (S2 Fig) |
| | Self-management triage assessment which gives patients advice based on symptoms (S1 Fig) |
| Monitoring | Paired blood pressure and heart rate monitor automatically stores measurements in cloud database for patient to view history and trends |
| | Patient's care team can monitor patients remotely and be alerted of unsatisfactory measurements |

application. A brief tutorial was conducted by Philips to familiarize the patient with blood pressure measurement and the functions of the device and Motiva tool, which was delivered directly to the patient's homes. The patients were allowed 6 weeks to explore and use SETAF on their own, based on instructions provided on how to use the device. After the end of the 6-week period, a time was arranged to collect back the tablet and device from the user, at which point an in-depth face to face interview was conducted to understand the user's experience based on a semi-structured questionnaire.

Participants were recruited between February-March 2018 via purposive sampling from the outpatient clinic of a heart center of a hospital in Singapore for this study. English speaking patients with stable AF condition aged 21 years and above were eligible for the study. Stable AF was defined as no cardiovascular hospitalization and having no change to the patient's medication or therapy in the past 3 months. Participants were approached at the outpatient clinic and asked if they would like to take part in the study, with the study explained to them. Consent was taken and a contact number was obtained from the patient to arrange for a suitable time for the blood pressure monitor and tablet to be delivered to the patient.

## Data collection

After using SETAF for 6 weeks, a single session, face-to-face, individual, semi-structured in-depth interview was carried out in the participant's homes by a trained research staff (AC), who was part of the research team. The research staff (AC) followed a flexible interview guide which encouraged participants to talk at length about their experiences and acceptability of the device and its features. All interviews were audio-recorded with the participant's consent and each interview lasted on average 30 minutes (min. 15 and max. 45 minutes). In addition, the patients were asked to complete a short 4-item survey after the semi-structured interview to measure frequency of use, usefulness of device, desire to continue using the device and willingness to pay for the device. The full interview guide and 4-item survey are included as S1 and S3 Files.

## Ethical considerations

The Institutional Review Board approved all aspects of the study (National Healthcare Group Domain Specific Review Board reference: 2017/00068). The study conforms with the principles outlined in the Declaration of Helsinki [18]. Written informed consent was obtained for all study participants.

## Data analysis

All interviews were transcribed verbatim by a research staff. Analysis was done inductively by the authors (JN & BW), following a thematic analysis approach as described by Braun and Clarke [19]. This approach was preferred to allow identification of themes and subthemes linked to the dataset. The authors familiarized themselves with the data by listening to the audio recordings repeatedly and reading the interview transcripts (data immersion). Following that, interview transcripts were reviewed line by line to generate the initial codes across the entire data corpus (code generation). Constant comparative analysis of interview transcripts and codes helped to refine the emergent codes which were then grouped into categories to identify themes and subthemes (searching for themes). Discussions between the co-researchers (JN, BW & LTW) took place frequently to compare interpretations and further define and refine themes and subthemes until a consensus was reached. Nvivo 12 (QSR International. Burlington, MA.) software was used for qualitative analyses.

The results from the short 4-item survey were analyzed descriptively. Percentages (%) were used to summarize frequency of use, usefulness of device and desire to continue using the device. Willingness to pay for the device was presented as a mean value together with the minimum and maximum range.

Application usage data were presented as percentage of participants who watched the educational videos and read the motivational messages. The total number of vital signs monitoring logged for each patient was divided by the duration of study (42 days) to calculate average number of vital signs monitoring done in a day for each patient. The completion rate of the self-management triage assessments was categorized into less than or equal to 25%, between 26 to 50%, between 51 to 75% and between 76% to 100%.

The findings from both the qualitative and quantitative data were then organized and presented according to the TAM.

### Rigour

Several measures were taken to adhere to the criteria of trustworthiness (credibility, transferability, dependability and confirmability) [20]. The interviewer established rapport with the participants and prolonged engagement during the interview to ensure credibility. Transcripts were cross-checked by the authors who did not take part in the transcription process. Two authors were actively involved in the coding process to allow for data triangulation and an audit trail was maintained during the data analysis. In addition, the collection of multiple sources of data allowed for data triangulation between qualitative and quantitative findings. Descriptions of the participant characteristics, context and verbatim quotes were provided in this paper to allow readers to exercise judgement on the transferability of findings.

## Results

### Demographic profile

A total of 138 participants were approached to participate in this study. Of these, 37 were enrolled in the study while the remaining declined participation citing reasons such as time constraints, finding it too troublesome, not comfortable with using technological gadgets and not being able to speak or read English. Of the 37 participants, 19 were interviewed. 4 of the participants who were not interviewed declined to complete the 4-item survey, however, their application usage data were still included in the analysis. The participant characteristics are summarized in Table 2. Majority of them were male and of Chinese ethnicity. At the time of the study, they had been diagnosed with AF for 1–25 years.

### The technology acceptance model

Fig 1 illustrates the qualitative and quantitative findings according to the basis of TAM.

### Perceived usefulness

**Improve AF knowledge.**   Many of the participants acknowledged that they previously lacked understanding about their own health condition. Some of them attributed this to insufficient opportunities for patient education during their consultation with the healthcare provider while others expressed that they are often skeptical of the information available on the internet. With SETAF they were able to access validated educational content and learn more about AF during their own time.

**Table 2. Participant characteristics.**

| Demographic Profile | Overall (n = 37) | Interviewed (n = 19) |
| --- | --- | --- |
| **Ethnicity** | | |
| Chinese | 30 | 14 |
| Malay | 7 | 5 |
| **Gender** | | |
| Male | 29 | 15 |
| Female | 8 | 4 |
| **Age** | | |
| **Mean** | 65.1 | 61.6 |
| **Range** | 41–78 | 43–77 |
| **Years diagnosed with AF** | | |
| Mean | 7.2 | 9.7 |
| Range | 1–25 | 1–25 |

"*Sometimes we want to ask so many questions when we go to the hospital, but sometimes it doesn't come out you know, like never think you know. But when we go back and forget to ask the doctor about this and that, at least with this one [SETAF] it gives me knowledge. At least I know.*"

*–Participant 1*

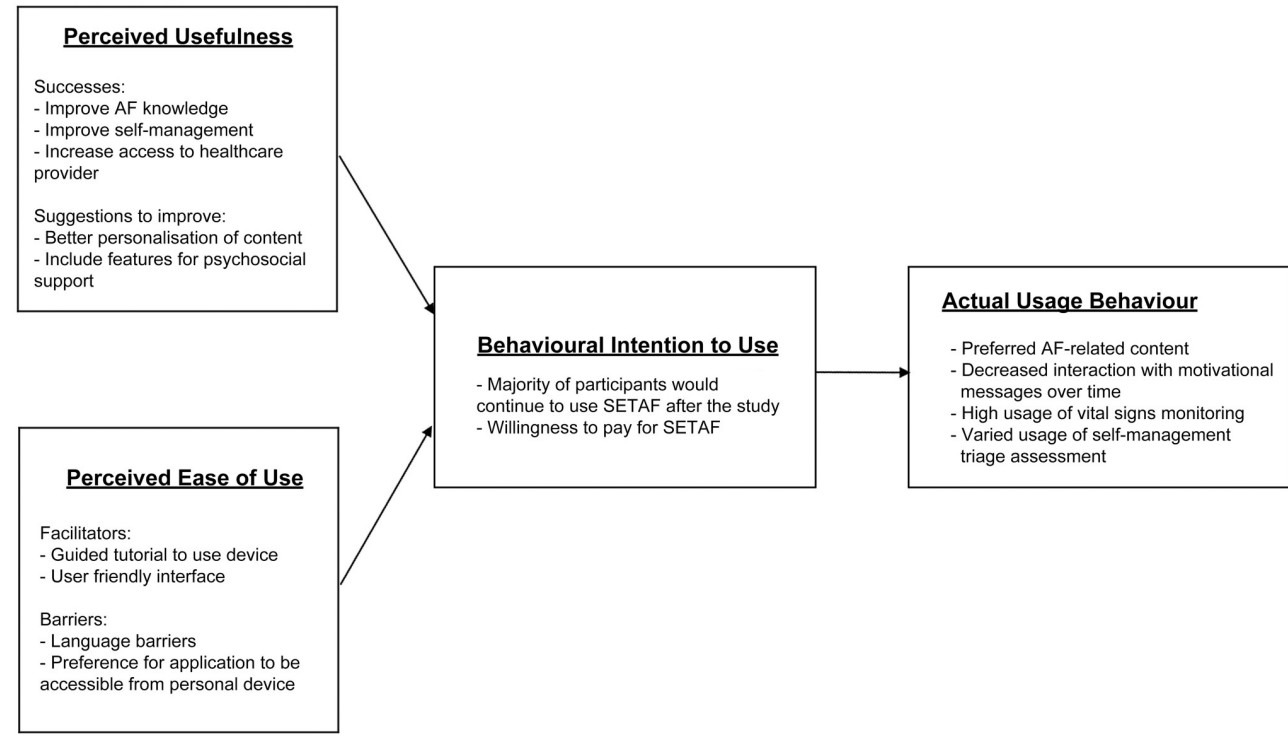

**Fig 1. Technology acceptance model for SETAF.**

Nearly all participants reported that SETAF helped them to understand more about AF as the videos allowed them to visualise the pathophysiology better. This in turn led them to be more aware of the risks associated with AF and the importance of treatment adherence.

"*I focused more on the stroke-wise, how AF works, if I never control my AF then the blood clogs. So now I know why the doctor is introducing blood thinner in order to safeguard me from getting a stroke, because the risk is there. From the video I could instantly know why, why do they do that, why they introduced [blood thinning medication]. Because I don't want to be on a lot of medications but now I know why.*"

*–Participant 11*

**Improve self-management.**   As part of the self-management algorithm, the SETAF application periodically sends reminders to reinforce healthy lifestyle habits such as exercising and keeping a balanced diet. If applicable, it will also provide actionable feedback in response to the blood pressure and heart rate measurements. Many participants felt that this helped them to stay motivated and was a source of encouragement to sustain healthier behaviours. They also reported feeling happy and that it boosted their confidence when they received positive feedback about their vital sign measurements.

"*Yes, definitely it reinforced [healthy lifestyle]. Like hey, you have to be careful you are this category. It helped to remind me like please continue to watch your diet. Please continue make sure to watch your sleep, you know get enough sleep. Please continue to watch your exercise, make sure you do it. These are very good timely reminders, there is a subtle reinforcement or reminder. It's good, it's very refreshing to see that there is a deliberate attempt by the application to give positive feedback in order to motivate you to keep your lifestyle.*"

*–Participant 9*

All the participants agreed that they preferred using SETAF to keep track of their blood pressure and heart rate measurements compared to the traditional paper-based recording as it is more convenient. Patients liked that they no longer needed to manually write down their measurements and felt that this was more organized than recording on paper. Some even expressed improved adherence as it prevents patients from 'cheating' and false reporting.

"*Paper is not advisable because too many things then you need to record, instead of app faster everything goes there. I don't need to care what is the thing just linked to there. I cannot do any cheating or cannot do any false report also, it's [blood pressure monitor] directly linked to there [SETAF application]. If say write one I can write whatever thing I preferred one.*"

*–Participant 6*

**Increase access to healthcare provider.**   SETAF uploads the blood pressure and heart rate measurements into a cloud database that is accessible remotely by the patient's care team and also sends alerts for unsatisfactory readings. Participants felt reassured that they were being monitored remotely and expressed that having a direct helpline for medical advice during symptomatic episodes can help to prevent unnecessary hospital admissions.

"*Because AF is sudden one quite serious one, how can you wait until next day? My only worry is like initial year that few year, I always have palpitation at night one about 10, 11 then very fast. So I was a bit worried whether I should go to the Accident and Emergency Department or*

*whether I should keep calm and you know? I also don't want to waste the manpower and all these a bit only you go A and E."*

*–Participant 29*

**Better personalization of content.** Many of the participants expressed different preferences for the educational content. While some found the lifestyle related videos which contained information about recommended exercises and diet useful, others felt that the information was too basic as they had received similar education during their counselling sessions with the dietician or other healthcare providers. Instead, they expressed more interest in learning about AF as it was more relevant to them.

*"Yah especially the exercise [videos] yah it doesn't interest me that much because I already know. Maybe like more on the AF then it will attract me more."*

*–Participant 11*

Some participants also preferred having more information on other age-related diseases as they had co-existing chronic conditions such as diabetes and hypertension and wanted to know how to better manage their health. Others who did not have these conditions expressed similar desire to learn more about other common diseases in order to prevent them.

*"I think I can speak only on behalf of the people of my age group, for elderly people. Any information on age related diseases will be most helpful. Age related disease in other words there are certain standard disease which we should prevent. How do we prevent from getting it or at least we are early notified then we know how to minimize the impact you see."*

*–Participant 3*

Another issue raised by some of the participants was that the motivational messages and reminders began to feel automated after a while. The lack of personal touch to the messages made them less willing to read and engage with the application as it felt repetitive.

*"I find because when you realized it's automated then you don't want to read. You don't want to read that motivation kind of thing. . . It's like oh well just another one. . . You will find that it's not so motivating anymore."*

*–Participant 2*

**Include features to provide psychosocial support.** Some participants suggested the inclusion of features to provide psychosocial support. They felt that living with AF can be emotionally stressful. However, there is a dearth of resources and information to help them cope with this aspect of the disease.

*"Maybe videos on how to overcome emotions because I don't see any coming from that viewpoint. Its more on introducing what is the diagnosis the medical term whatever and the healthy cooking diet and even exercise but nothing touched on the emotion side. Yah so that people can understand because when someone changed from behaviour or whatever usually suspect is from that sickness but why? Why it happen and what triggered it, then how to overcome it. I think that is very useful because when I tried to find out about this in the web there's*

*not much things also touching on how to. . . they never teach like how to cope with stress, how to cope with changes in your lifestyle.*"

*–Participant 11*

They also recalled feeling nervous and overwhelmed when they were first diagnosed with AF, especially during symptomatic episodes. They attributed this to lack of knowledge and misconceptions about the disease and highlighted the value of peer support as a source of encouragement and reassurance.

"*Maybe you all can set a what is that called, a group that have AF support. You see some of them might be nervous don't know what to do, just encourage all of them, need encouragement and support group I say.*"

*–Participant 8*

## Perceived ease of use

**Facilitators of technology update.** The participants in this study had varying levels of experience with using technological devices. However, almost all agreed that SETAF was easy to use and did not encounter any technical difficulties. A key component that helped the participants feel comfortable using SETAF was the tutorial given at the beginning of the study to help familiarise them with the device and application. The participants with less exposure to previous technology use reported that this helped them to overcome the barrier towards using SETAF as they would have otherwise assumed it was complicated or difficult to use.

"*I think overall, most of them [features] are very easy. . .All of them are very easy to manage because when your colleague deliver of course she explained to me but still you have to go in you know to try it. I was a bit apprehensive in the beginning when I first started using it but then I realized that it's very straight forward and very easy to use.*"

*–Participant 4*

**Reduce language barriers.** Some participants reported difficulty understanding certain words or terms used. They recalled having to look up the meaning of these words on the internet. One participant did not attempt the surveys at all as she did not understand them.

"*Some of the words are really a bit a bit chim [Singaporean slang meaning "difficult to understand"] so yah I don't understand. . .*"

*–Participant 24*

Another usability concern raised by some participants was the language of the intervention, which was only available in English. They felt that the older generation in Singapore may only be literate in their own mother tongue such as Mandarin and would not be able to use SETAF.

"*Those like the old people they are Chinese, they want to read Chinese so they are able to understand. If not then they only give to those people who said they must understand English. The Chinese educated, Chinese people only can read Chinese.*"

*–Participant 6*

**Preference for application to be accessible from personal device.** The general preference of the participants was to access SETAF with their own personal devices. While they found SETAF useful in helping them manage their condition, they expressed desire to minimize owning multiple devices and were more willing to continue using it if it was an application that they can use on their own phones or computers.

> "If you asked me, phone. Because it's with you every time 24–7 right and if they can move forward without issuing a tablet to the patient because I think inconvenience whatever right, I think just send them a link, user ID and password ask them to go internet. . ."

> —Participant 11

## Behavioral intention to use

Of the 33 participants who completed the survey, 94% (31/33) reported using SETAF almost every day (6–7 days a week), while the remaining 6% (2/33) used it frequently (4–5 days a week). Of participants, 21% (7/33) strongly agreed, 73% (24/32) agreed and 6% (2/33) were neutral that SETAF was useful in helping them manage their AF condition. When asked if they would continue using SETAF after the study, 21% (7/33) strongly agreed, 49% (16/33) agreed, 24% (8/33) were neutral and 6% (2/33) disagreed. The participants were willing to pay an average of $112.08 SGD (range $10–500) to purchase SETAF to continue using it.

## Actual use behavior

Seven educational videos of varying topics were included in the application. The length of the videos ranged from approximately 2 to 25 minutes. The patient interactions with the videos are summarized in Fig 2. The most watched videos were titled 'Atrial Fibrillation Explained' (91.9%) and 'Clot Formation and Stroke Risks' (78.4%) while the least watched video was 'Cook Right and Eat Smart for Healthy Ageing' (56.8%).

Motivational messages were generated from the message bank and sent to the participants at regular intervals of approximately 2 messages per week during the study. The patient interactions with messages from the application are summarized in Fig 3. The percentage of patients who read the messages decreased over time from 94.6% for the first motivational message to 67.6% for the last motivational message.

Table 3 summarizes the participants' usage of SETAF's blood pressure monitor and self-management triage assessment. On average, most of the participants used SETAF to measure their blood pressure at least once a day (75.7%). The remaining 24.3% of the participants used SETAF to measure blood pressure less than once a day during the study. The self-management triage assessment was pushed out to the participants every day during the study. However, the completion rates varied greatly between participants. 35.1% of the participants completed less than 25% of the self-management triage assessments and 45.9% of the participants completed more than 75% of the assessments.

## Discussion

### Principal findings

To our knowledge, this is the first qualitative study in Singapore to explore the acceptability and usability of an mHealth intervention for AF patients. Acceptance of SETAF relied on its potential to fill the gaps in current AF care while being easy and convenient for patients to use. In particular, they perceived the patient education and self-monitoring functions as useful for

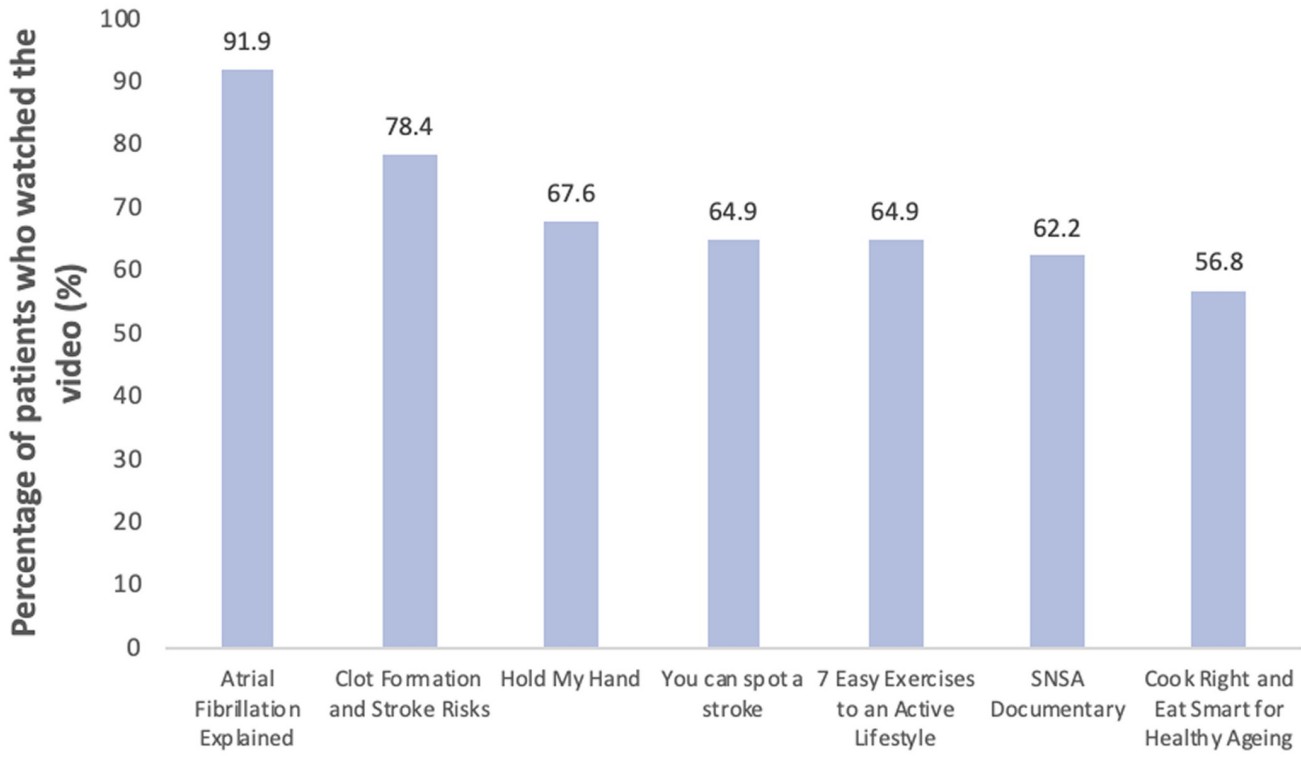

**Fig 2. Patient interactions with educational videos.**

managing AF. They were also reassured by the link the app provided to their healthcare providers. This perceived usefulness is reflected in their high usage of SETAF, with all of the participants reporting using it at least 4–5 days a week Moreover, more than half of the participants would continue to use SETAF and were willing to pay for it. Interestingly, the SETAF application usage data showed that participants were more likely to interact with AF-related content, while interaction with the motivational messages declined over time. Our study also identified suggestions which could improve the acceptance of SETAF, including: better personalization of content, the need for psychosocial support, reducing language barriers and accessibility of SETAF. These findings support the acceptance of mHealth technology amongst the majority of our AF patients and their insights can inform the design of future mHealth interventions for them. For those patients who were unwilling to use SETAF in the future, an improved version of SETAF could potentially change their opinion.

## Perceived usefulness: Improved patient education

Education is an essential component of care as patients' knowledge is a known determinant of therapy uptake and adherence [21–23]. However, previous studies found that AF patients had poor knowledge of their condition and therapy [21–24]. Clinicians were cited as their main source of information and they supplemented this with independent internet searches, which had limitations such as potentially leading to inaccurate and misleading information [23].

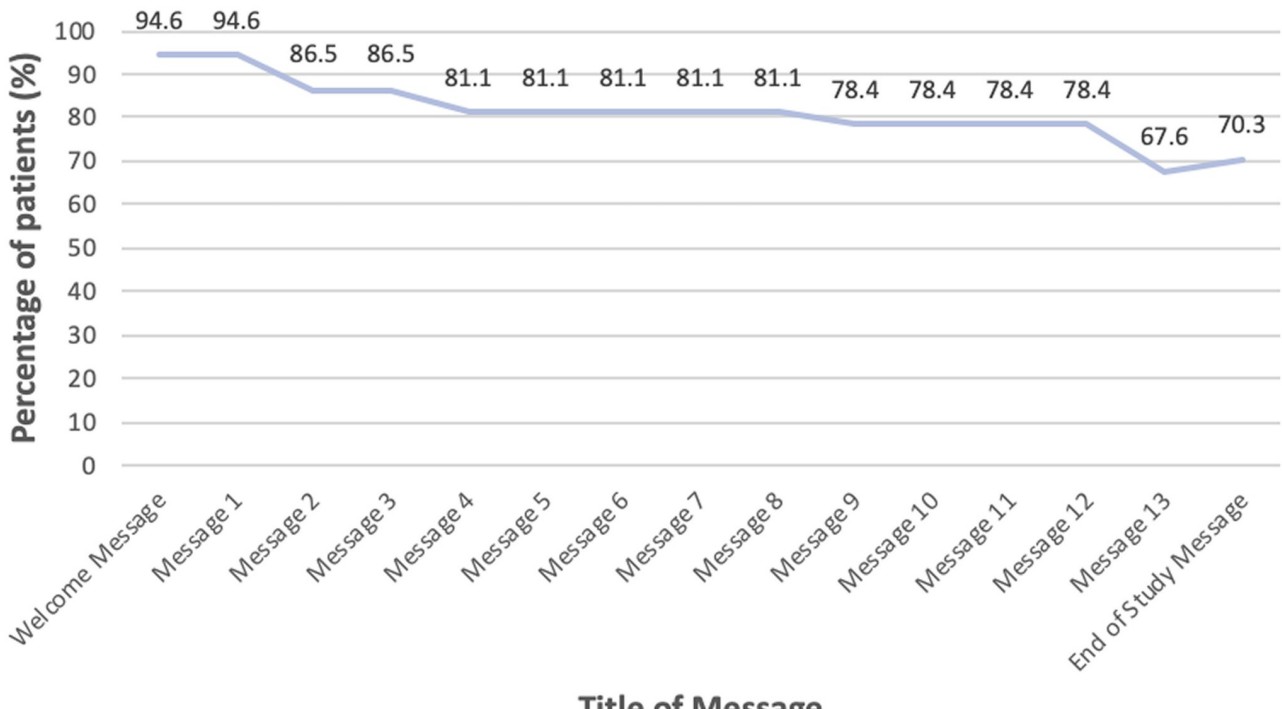

**Fig 3. Patient interactions with motivational messages.**

Similarly, our study provided evidence that reliance on healthcare professionals and the internet for patient education continued to be suboptimal for our patients. Many participants in the present study felt they had limited opportunities to learn clinical relevant knowledge, which was imparted within the constraints of a clinic consultation which resulted in inadequate knowledge at baseline. However, they reported increased knowledge of AF after using the SETAF application, which in turn led to improved perceptions of the necessity of their treatment. Thus, mHealth has the potential to serve as a supplementary educational tool for delivering curated and validated content to increase health literacy.

**Table 3. Use of blood pressure monitor and self-management triage assessment.**

|  | No of participants (n = 37) | Percentage (%) |
|---|---|---|
| **Frequency of blood pressure monitor use** | | |
| **Less than once a day** | 9 | 24.3 |
| **Once a day** | 20 | 54.1 |
| **Twice a day** | 7 | 18.9 |
| **Thrice a day** | 1 | 2.7 |
| **Completion rate of Self-management Triage Assessment** | | |
| **≤25%** | 13 | 35.1 |
| **26–50%** | 4 | 10.8 |
| **51–75%** | 3 | 8.1 |
| **76–100%** | 17 | 45.9 |

## Perceived ease of use: Language barriers

On the other hand, reading ability also plays a vital role in health literacy [11]. mHealth interventions should not only ensure scientifically validated content but also be tailored to the literacy levels of their users to maximize its benefits [11, 25]. This is further emphasized by the findings in our subtheme to reduce language barriers. Participants recalled encountering difficult terms which they had to search for. One participant even reported not attempting the surveys that were part of the education component of SETAF due to limited comprehension. AF patients in Singapore tend to be older with lower levels of formal education [13]. Given the multiethnic groups in Singapore, the lack of language options was also considered to be a shortcoming. These considerations should be taken into account when developing the educational content of mHealth interventions.

## Perceived usefulness: Improved self-management

During the development phase of SETAF, vital signs monitoring and access to advice from healthcare professionals were identified as desirable features for the mHealth tool [13]. Hence, a paired blood pressure monitor was included in the intervention to allow patients to measure their heart rate and blood pressure. These measurements were then automatically uploaded into the cloud database and made accessible to both patients and their healthcare team. From the SETAF usage data, we found that most patients used the blood pressure monitoring function at least once a day. The study participants felt that this was a better way to keep track of their vital signs compared to traditional paper and pen methods and likely improved adherence to self-monitoring.

SETAF included a self-management triage assessment algorithm which would provide patients with actionable feedback based on the severity of their symptoms such as palpitations, chest pain and shortness of breath. The programmed decision algorithm may direct patients to call the hotline for medical advice for low-risk patients or to seek help immediately by calling an ambulance for emergent cases. We found only modest usage of the self-management triage assessment, and less than half of the participants completed 75% or more of the triage assessments. However, during the interview, most of the patients were aware of this feature and the hotline available to contact for medical advice. When prompted, the participants did not use the hotline as they felt well but found it to be useful and thought it could potentially reduce unnecessary hospital visits during symptomatic episodes. In addition, participants felt reassured that their care team would be notified of abnormal readings from the blood pressure and heart rate monitor. This sense of reassurance from mHealth monitoring was also observed in a qualitative study exploring patients' experiences of using an mHealth application for management of Chronic Obstructive Pulmonary Disease [26]. Patients with AF are frequent users of the emergency department due to the sporadic and unpredictable nature of the disease [10]. Furthermore, traditional ambulatory care settings provide only a snapshot of the patient's AF condition and the reliance on patient-reported symptoms makes it impossible to detect periods of silent AF which augment stroke risk and cardiomyopathy from untreated tachycardia [12]. Aljuaid et al. demonstrated the feasibility of smartphone electrocardiogram (ECG) monitoring to lower emergency room and clinic visits for post-ablation AF patients [27]. Similarly, the Mobile Atrial Fibrillation App (mAFA) trial which includes cardiac rhythm monitoring and two-way communication with doctors found reduced risk of adverse clinical events and rehospitalization [28–31]. Such technological advancements of mHealth show promise to transform the delivery and quality of AF care.

## Perceived usefulness: Tailoring and customization features

In recent years, text message based mHealth interventions have gained popularity as a potential way to modify health behaviors. Pooled evidence consistently shows that text messaging interventions have a positive effect in health promotion [28, 30]. Interestingly, while many participants in the present study found the daily reminders from SETAF to be useful and helped to reinforce lifestyle changes, some thought that the messages became repetitive and lost effectiveness over time. In addition, the application usage data showed a decreasing trend in percentage of patients who read the motivational messages over time. These findings support the notion that targeted, tailored and personalized messages are more engaging and increases intervention efficacy [28]. Targeted messages cater the content to a population sub-group based on shared group characteristics, tailored messages are unique to an individual's assessed characteristics and personalized messages included the participant's name. In addition, messaging frequency was also found to moderate the intervention effectiveness [32]. Interventions that allowed users to customize the frequency and timing of the messages presumably led to the messages being delivered optimally to individual's needs. The use of tailoring strategies is thought to enhance message processing and acceptance by the users and consequently improve the intervention efficacy [23, 33]. The optimal strategies for this intervention are unclear and will need further study.

## Perceived usefulness; addressing psychosocial needs

Another key finding was the need for psychosocial support. Patient anxiety and depression is a well-known issue among AF patients and has recently been identified as an education need [34]. In the present study, participants discerned lack of knowledge as a contributing factor to their anxiety and suggested that the addition of emotional education and peer support could help to reduce it. This is congruent with current evidence that education has a small but positive effect on anxiety and depression for AF patients [8]. In addition, the European Society of Cardiology guideline has underscored the role of psychosocial management in integrated AF care [11]. Current mHealth interventions for AF patients are largely focused on patient education, medication adherence and self-monitoring [13]. To our knowledge, no mHealth interventions have addressed the psychosocial needs of AF patients. Future mHealth interventions for AF management should consider the addition of emotional education and other features of psychosocial support for a more holistic approach.

## Strengths and limitations of study

A strength of our study was the conduct of in-depth interviews based on actual trial use of an mHealth tool. Other studies exploring the acceptance and use of mHealth tools so far are based on expressed intention to use and not actual usage. In addition, the qualitative data was supplemented by quantitative data from the survey and application usage data. This allowed for data triangulation between the different data sets and gave us a deeper insight into the participant's usage behavior. Our sample is also ethnically diverse and includes patients with a wide range of years living with AF. This reflects the different needs and preferences of newly diagnosed and chronic AF patients. Of note, the mHealth tool used in this study was designed based on the findings of a previous study by Cher et al. [13]. Tailoring SETAF to the needs and preferences of patients in the local context which were identified in the previous study may have helped to achieve the perceived usefulness, ease of use and acceptance by the patients in this study.

However, the findings of this study should be considered in the context of a few limitations. Firstly, due to the limitations of the mHealth tool, which was only available in English, we

were not able to capture the perspectives of other non-English speaking AF patients who may have specific preferences and mobile health technology use behaviors. Secondly, improvements in AF knowledge and self-management were self-reported during the interviews and no objective measurements were taken to confirm the participants claims. Nonetheless, the purpose of this study was not to assess the effectiveness of the mHealth tool but to explore its acceptability and usefulness for AF patients.

## Conclusion

Our patients found SETAF to be an acceptable tool for the management of AF, and they particularly valued its convenience, educational and self-monitoring functions. Patients' insights on features they found useful and suggestions for further improvement should guide the design and implementation of future mHealth interventions for AF. The findings from this study suggests that an opportunity exists to improve AF care by demonstrating the acceptability and usefulness of implementing an mHealth tool tailored for AF patients in Singapore.

## Supporting information

**S1 Fig. Self-management triage decision algorithm.**
(TIF)

**S2 Fig. Examples of motivational messages.**
(TIF)

**S1 File. 4-item survey.**
(DOCX)

**S2 File. Standards for reporting qualitative research checklist.**
(DOCX)

**S3 File. Interview guide.**
(DOCX)

**S1 Data.**
(ZIP)

## Acknowledgments

We would like to thank Koninklijke Philips N.V. for loaning the tablets. We also thank Dr. Joanne Yoong and Dr. Luo Nan for their input in the study methodology. Lastly, we would like to thank the patients for their enthusiastic participation in the study.

## Author Contributions

**Conceptualization:** Jennifer Nathania, Brigitte Fong Yeong Woo, Boon Piang Cher, Wei-Yan Aloysius Chia, Yee Wei Lim, Hubertus J. M. Vrijhoef, Toon Wei Lim.

**Formal analysis:** Jennifer Nathania, Brigitte Fong Yeong Woo, Toon Wei Lim.

**Funding acquisition:** Toon Wei Lim.

**Investigation:** Boon Piang Cher, Kai Yee Toh, Wei-Yan Aloysius Chia.

**Methodology:** Wei-Yan Aloysius Chia.

**Project administration:** Jennifer Nathania, Boon Piang Cher, Kai Yee Toh, Wei-Yan Aloysius Chia.

**Supervision:** Yee Wei Lim, Hubertus J. M. Vrijhoef, Toon Wei Lim.

**Writing – original draft:** Jennifer Nathania, Brigitte Fong Yeong Woo.

**Writing – review & editing:** Boon Piang Cher, Kai Yee Toh, Wei-Yan Aloysius Chia, Yee Wei Lim, Hubertus J. M. Vrijhoef, Toon Wei Lim.

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
