## [Decision Letter · Decision Letter 0]

6 Sep 2021

PONE-D-21-19770Patient perspectives about mobile health to improve atrial fibrillation self-management and education: a mixed-methods study in SingaporePLOS ONE

Dear Dr. Woo,

Thank you for submitting your manuscript to PLOS ONE. After careful consideration, we feel that it has merit but does not fully meet PLOS ONE’s publication criteria as it currently stands. Therefore, we invite you to submit a revised version of the manuscript that addresses the points raised during the review process.

We look forward to receiving your revised manuscript.

Kind regards,

Muhammad Junaid Farrukh

Academic Editor

PLOS ONE

Journal Requirements:

2. Please include additional information regarding the survey or questionnaire used in the study and ensure that you have provided sufficient details that others could replicate the analyses. For instance, if you developed a questionnaire as part of this study and it is not under a copyright more restrictive than CC-BY, please include a copy, in both the original language and English, as Supporting Information

3. When reporting the results of qualitative research, we suggest consulting the COREQ guidelines  or other relevant checklists listed by the Equator Network, such as the SRQR, to ensure complete reporting (http://journals.plos.org/plosone/s/submission-guidelines#loc-qualitative-research). Moreover, please provide the interview guide used as a Supplementary File.

4. Please provide additional details regarding participant consent. In the ethics statement in the Methods and online submission information, please ensure that you have specified (1) whether consent was informed and (2) what type you obtained (for instance, written or verbal, and if verbal, how it was documented and witnessed). If your study included minors, state whether you obtained consent from parents or guardians. If the need for consent was waived by the ethics committee, please include this information.

For additional information about PLOS ONE ethical requirements for human subjects research, please refer to " ext-link-type="uri" xlink:type="simple">http://journals.plos.org/plosone/s/submission-guidelines#loc-human-subjects-research.")( you can modify it if you want, just leaving the part about retrospective studies)

5. Please ensure that you include a title page within your main document. We do appreciate that you have a title page document uploaded as a separate file, however, as per our author guidelines (http://journals.plos.org/plosone/s/submission-guidelines#loc-title-page) we do require this to be part of the manuscript file itself and not uploaded separately.

Could you therefore please include the title page into the beginning of your manuscript file itself, listing all authors and affiliations

"This study was funded by a Health Services Research New Investigator Grant (HSRNIG13nov002) from the National Medical Research Council, Singapore. We would like to thank Koninklijke Philips N.V. for loaning the tablets. Neither the funders nor Philips have any role in the study design, data collection, analysis and preparation of manuscript. We also thank Dr. Joanne Yoong and Dr. Luo Nan for their input in the study methodology. Lastly, we would like to thank the patients for their enthusiastic participation in the study."

"This study was funded by a Health Services Research New Investigator Grant (HSRNIG13nov002) from the National Medical Research Council, Singapore."

7.We note that you have stated that you will provide repository information for your data at acceptance. Should your manuscript be accepted for publication, we will hold it until you provide the relevant accession numbers or DOIs necessary to access your data. If you wish to make changes to your Data Availability statement, please describe these changes in your cover letter and we will update your Data Availability statement to reflect the information you provide.

Reviewers' comments:

Reviewer's Responses to Questions

**Comments to the Author**

1. Is the manuscript technically sound, and do the data support the conclusions?

Reviewer #1: Partly

Reviewer #2: Partly

2. Has the statistical analysis been performed appropriately and rigorously? 

Reviewer #1: Yes

Reviewer #2: No

3. Have the authors made all data underlying the findings in their manuscript fully available?

Reviewer #1: Yes

Reviewer #2: Yes

4. Is the manuscript presented in an intelligible fashion and written in standard English?

Reviewer #1: Yes

Reviewer #2: Yes

5. Review Comments to the Author

Reviewer #1: Dear Authors,

Please kindly consider the following suggestions for further improvement of the manuscript:

1. Title: To change the term 'mobile health' to the actual tool used i.e. SETAF

2. Methods: To include any specific info or features of SETAF that patients are required to explore during the 6-week duration. My concern is some patients may not explore at all or focus only 1-2 features that they like the most, given that there is a huge gap in the age of patients participated in this study. This is of course, unless, the researcher could remotely track all patients' use of SETAF throughout the duration.

3. Methods: To include a more detailed inclusion and exclusion criteria for the patients. For example, background educational level, familiarity with using smart phones and apps etc. This could potentially affect the results of this study.

4. Data analysis: To correct the completion rate of the self-management triage assessment. I noticed there is an overlap between the values i.e., 25-50% and 50-75%. The info in the table 3 is correct.

5. Data collection: To include the detail on whether or not a consent is obtained prior to the audio-recording.

6. Results: There were different N used in the results section. In Line 365, it was mentioned that 33 participated in the survey, and not 38 as mentioned in the earlier section. Please clarify.

Reviewer #2: Participants ages were stated by the author begins from 25 years old , while the

mean mean age was nearly 65 years old. This age issue represent an acceptable conflict as this high variation in participants ages implies different motivations to use mobile health . The validity and reliability of the used mobile health wasn't mentioned in the study . The author didn't mention how intra co- researcher variability was overcomed. Regarding to sample size calculation was not mentioned , however sample size was small. statistical method was unclear and not obviously mentioned.

6. PLOS authors have the option to publish the peer review history of their article (what does this mean?). If published, this will include your full peer review and any attached files.

Reviewer #1: No

Reviewer #2: No

---

## [Author Response · Author response to Decision Letter 0]

13 Oct 2021

Editor’s comments to author

https://imsva91-ctp.trendmicro.com:443/wis/clicktime/v1/query?url=https%3a%2f%2fjournals.plos.org%2fplosone%2fs%2ffile%3fid%3dwjVg%2fPLOSOne%5fformatting%5fsample%5fmain%5fbody.pdfumid=04E8D6E5-CB5D-1405-9D93-56F7C9BF425Cauth=6e3fe59570831a389716849e93b5d483c90c3fe4-555acbf8fc5d83da67da269c4cb2d79d882a2059 and 

https://imsva91-ctp.trendmicro.com:443/wis/clicktime/v1/query?url=https%3a%2f%2fjournals.plos.org%2fplosone%2fs%2ffile%3fid%3dba62%2fPLOSOne%5fformatting%5fsample%5ftitle%5fauthors%5faffiliations.pdfumid=04E8D6E5-CB5D-1405-9D93-56F7C9BF425Cauth=6e3fe59570831a389716849e93b5d483c90c3fe4-8c47293dd5dd6502c7a8c7993d1ba4a88f1d6711

Response: The manuscript has been revised to meet PLOS ONE’s style requirements

2. Please include additional information regarding the survey or questionnaire used in the study and ensure that you have provided sufficient details that others could replicate the analyses. For instance, if you developed a questionnaire as part of this study and it is not under a copyright more restrictive than CC-BY, please include a copy, in both the original language and English, as Supporting Information

Response: The survey used in this questionnaire has been included as Supporting Information (S1. File)

3. When reporting the results of qualitative research, we suggest consulting the COREQ guidelines or other relevant checklists listed by the Equator Network, such as the SRQR, to ensure complete reporting (https://imsva91-ctp.trendmicro.com:443/wis/clicktime/v1/query?url=http%3a%2f%2fjournals.plos.org%2fplosone%2fs%2fsubmission%2dguidelines%23loc%2dqualitative%2dresearchumid=04E8D6E5-CB5D-1405-9D93-56F7C9BF425Cauth=6e3fe59570831a389716849e93b5d483c90c3fe4-860c02cb43b02a39d38719c1b99d0fb90a836b38). Moreover, please provide the interview guide used as a Supplementary File.

Response: SRQR was used in the reporting of this study as mentioned in line 115 and the completed checklist is submitted as Supporting information (S2 File). The interview guide is also included as Supporting information (S2 File).

4. Please provide additional details regarding participant consent. In the ethics statement in the Methods and online submission information, please ensure that you have specified (1) whether consent was informed and (2) what type you obtained (for instance, written or verbal, and if verbal, how it was documented and witnessed). If your study included minors, state whether you obtained consent from parents or guardians. If the need for consent was waived by the ethics committee, please include this information.

For additional information about PLOS ONE ethical requirements for human subjects research, please refer to https://imsva91-ctp.trendmicro.com:443/wis/clicktime/v1/query?url=http%3a%2f%2fjournals.plos.org%2fplosone%2fs%2fsubmission%2dguidelines%23loc%2dhuman%2dsubjects%2dresearch.%22%29%28umid=04E8D6E5-CB5D-1405-9D93-56F7C9BF425Cauth=6e3fe59570831a389716849e93b5d483c90c3fe4-7648a745adfe6670965b5c7a2d02c4758e4d64b6 you can modify it if you want, just leaving the part about retrospective studies)

Response: We have further clarified that written informed consent was obtained for all study participants (line 163-164).

5. Please ensure that you include a title page within your main document. We do appreciate that you have a title page document uploaded as a separate file, however, as per our author guidelines (https://imsva91-ctp.trendmicro.com:443/wis/clicktime/v1/query?url=http%3a%2f%2fjournals.plos.org%2fplosone%2fs%2fsubmission%2dguidelines%23loc%2dtitle%2dpageumid=04E8D6E5-CB5D-1405-9D93-56F7C9BF425Cauth=6e3fe59570831a389716849e93b5d483c90c3fe4-82079791b1d8a8f6742f24408a79b447b3df4601) we do require this to be part of the manuscript file itself and not uploaded separately.

Could you therefore please include the title page into the beginning of your manuscript file itself, listing all authors and affiliations

Response: A title page listing all authors and affiliations has been included in the beginning of our manuscript file.

"This study was funded by a Health Services Research New Investigator Grant (HSRNIG13nov002) from the National Medical Research Council, Singapore. We would like to thank Koninklijke Philips N.V. for loaning the tablets. Neither the funders nor Philips have any role in the study design, data collection, analysis and preparation of manuscript. We also thank Dr. Joanne Yoong and Dr. Luo Nan for their input in the study methodology. Lastly, we would like to thank the patients for their enthusiastic participation in the study."

"This study was funded by a Health Services Research New Investigator Grant (HSRNIG13nov002) from the National Medical Research Council, Singapore."

Response: The statement ‘This study was funded by a Health Services Research New Investigator Grant (HSRNIG13nov002) from the National Medical Research Council, Singapore’ has been removed from our Acknowledgments section. 

7.We note that you have stated that you will provide repository information for your data at acceptance. Should your manuscript be accepted for publication, we will hold it until you provide the relevant accession numbers or DOIs necessary to access your data. If you wish to make changes to your Data Availability statement, please describe these changes in your cover letter and we will update your Data Availability statement to reflect the information you provide.

Response: No papers in the reference list have been retracted. No addition or removal was made to the reference list. References 1, 3 and 4 have been updated to reflect the latest available statistics. 

Reviewer’s comments to authors

Reviewer #1: Dear Authors,

Please kindly consider the following suggestions for further improvement of the manuscript:

1. Title: To change the term 'mobile health' to the actual tool used i.e. SETAF

Response: Thank you for your suggestion we have updated our title accordingly.

2. Methods: To include any specific info or features of SETAF that patients are required to explore during the 6-week duration. My concern is some patients may not explore at all or focus only 1-2 features that they like the most, given that there is a huge gap in the age of patients participated in this study. This is of course, unless, the researcher could remotely track all patients' use of SETAF throughout the duration.

Response: Information regarding the features of SETAF has been described in our Methods section under ‘Setting and sample’ (line 120-129) and summarised in Table 1. The patients were encouraged to explore all the features, however the research team did not dictate how they should do so in order to capture their actual usage behaviour. The patients’ actual usage behaviour was described in our Results section (line 382-405)

3. Methods: To include a more detailed inclusion and exclusion criteria for the patients. For example, background educational level, familiarity with using smart phones and apps etc. This could potentially affect the results of this study.

Response: The inclusion and exclusion criteria for the patients were intentionally broad to be able to capture the viewpoints of patients from diverse backgrounds and with varying experience using technology.

4. Data analysis: To correct the completion rate of the self-management triage assessment. I noticed there is an overlap between the values i.e., 25-50% and 50-75%. The info in the table 3 is correct.

Response: Thank you for your correction, we have amended line 189-191 to reflect this change.

5. Data collection: To include the detail on whether or not a consent is obtained prior to the audio-recording.

Response: We have added that audio-recording during the interviews were done with participant’s consent (line 156).

6. Results: There were different N used in the results section. In Line 365, it was mentioned that 33 participated in the survey, and not 38 as mentioned in the earlier section. Please clarify. 

Response: In total, 37 participants were recruited to participate in this study. 4 participants declined to complete the survey, hence only data from 33 participants were available. The demographics and application usage data from these 4 participants were still included in the data analysis. We have updated our Results section to clarify this (line 215-217).

Reviewer #2: Participants ages were stated by the author begins from 25 years old , while the

mean mean age was nearly 65 years old. This age issue represent an acceptable conflict as this high variation in participants ages implies different motivations to use mobile health . The validity and reliability of the used mobile health wasn't mentioned in the study . The author didn't mention how intra co- researcher variability was overcomed. Regarding to sample size calculation was not mentioned , however sample size was small. statistical method was unclear and not obviously mentioned.

Response: 

Although the initial inclusion criteria for the study was AF patients aged 21 years and above, the final age range of recruited participants was 41-78 years old as AF is more prevalent in the older population.

The validity and reliability of the SETAF were not reported in this study. However, these been explored and reported in an earlier study which contributed to the development of SETAF (Cher BP, Kembhavi G, Toh KY, Audimulam J, Chia W-YA, Vrijhoef HJ, et al. Understanding the Attitudes of Clinicians and Patients Toward a Self-Management eHealth Tool for Atrial Fibrillation: Qualitative Study. JMIR Human Factors [Internet]. 2020 Sep 17 [cited 2020 Oct 21];7(3):e15492.)

The interviews and surveys were done by the same co-author, thus minimizing variability during data collection. In addition, we explained that the co-authors discussed frequently to compare interpretations until consensus was reached for the themes and subthemes. (line 177-179, 201-202).

The primary objective of this study was to collect qualitative data on AF patients’ perspectives of SETAF, hence the sample was purposively selected and continued until data saturation was achieved. Participants who did not want to be interviewed were still recruited and data from the survey and application usage data were collected and included in the analysis. 

Due to the small sample size, we did not perform any statistical test and reported our findings descriptively (line 182-192).

---

## [Editor Report · Decision Letter 1]

16 Dec 2021

Patient perspectives of the Self-management and Educational Technology tool for Atrial Fibrillation (SETAF): a mixed-methods study in Singapore

PONE-D-21-19770R1

Dear Dr. %Brigitte Fong Yeong Woo%,

We’re pleased to inform you that your manuscript has been judged scientifically suitable for publication and will be formally accepted for publication once it meets all outstanding technical requirements.

Kind regards,

Muhammad Junaid Farrukh

Academic Editor

PLOS ONE

Additional Editor Comments (optional):

dear author, thank you for addressing all the comments by the reviewers. i am happy to say that your manuscript is now suitable for publication.
---

## [Editor Report · Acceptance letter]

31 Dec 2021

PONE-D-21-19770R1 

Patient perspectives of the Self-management and Educational Technology tool for Atrial Fibrillation (SETAF): a mixed-methods study in Singapore 

Dear Dr. Woo:

I'm pleased to inform you that your manuscript has been deemed suitable for publication in PLOS ONE. Congratulations! Your manuscript is now with our production department. 

Kind regards, 

on behalf of

Dr. Muhammad Junaid Farrukh 

Academic Editor

PLOS ONE